# Estimating effects of parents' cognitive and non-cognitive skills on offspring education using polygenic scores

Perline A. Demange [1,2,3] ✉, Jouke Jan Hottenga [1], Abdel Abdellaoui [4], Espen Moen Eilertsen [5,6], Margherita Malanchini [7,8], Benjamin W. Domingue [9,10,11], Emma Armstrong-Carter [9,11], Eveline L. de Zeeuw [1,3], Kaili Rimfeld [8,12], Dorret I. Boomsma [1], Elsje van Bergen [1,3], Gerome Breen [8,13], Michel G. Nivard [1] & Rosa Cheesman [5,8] ✉

Understanding how parents' cognitive and non-cognitive skills influence offspring education is essential for educational, family and economic policy. We use genetics (GWAS-by-subtraction) to assess a latent, broad non-cognitive skills dimension. To index parental effects controlling for genetic transmission, we estimate indirect parental genetic effects of polygenic scores on childhood and adulthood educational outcomes, using siblings ($N = 47,459$), adoptees ($N = 6407$), and parent-offspring trios ($N = 2534$) in three UK and Dutch cohorts. We find that parental cognitive and non-cognitive skills affect offspring education through their environment: on average across cohorts and designs, indirect genetic effects explain 36–40% of population polygenic score associations. However, indirect genetic effects are lower for achievement in the Dutch cohort, and for the adoption design. We identify potential causes of higher sibling- and trio-based estimates: prenatal indirect genetic effects, population stratification, and assortative mating. Our phenotype-agnostic, genetically sensitive approach has established overall environmental effects of parents' skills, facilitating future mechanistic work.

Parents and children tend to have similar educational outcomes[1]. Since education is highly predictive of social mobility and health across the lifespan[2,3], understanding the mechanisms underlying the intergenerational transmission of education could inform efforts to alleviate inequalities. Many studies have investigated how much certain parental characteristics influence offspring education, but relatively few have considered non-cognitive skills. The term 'non-cognitive' describes skills that differ from what has traditionally been

[1]Department of Biological Psychology, Vrije Universiteit Amsterdam, Amsterdam, The Netherlands. [2]Amsterdam Public Health Research Institute, Amsterdam University Medical Centers, Amsterdam, The Netherlands. [3]Research Institute LEARN!, Vrije Universiteit Amsterdam, Amsterdam, The Netherlands. [4]Department of Psychiatry, Amsterdam UMC, University of Amsterdam, Amsterdam, The Netherlands. [5]PROMENTA Research Center, Department of Psychology, University of Oslo, Oslo, Norway. [6]Centre for Fertility and Health, Norwegian Institute of Public Health, Oslo, Norway. [7]Department of Biological and Experimental Psychology, School of Biological and Chemical Sciences, Queen Mary University of London, London, UK. [8]Social, Genetic & Developmental Psychiatry Centre, Institute of Psychiatry, Psychology & Neuroscience, King's College London, London, UK. [9]Graduate School of Education, Stanford University, Stanford, CA, USA. [10]Center for Population Health Sciences, Stanford University, Stanford, CA, USA. [11]Center for Education Policy Analysis, Stanford University, Stanford, CA, USA. [12]Department of Psychology, Royal Holloway University of London, London, UK. [13]NIHR Maudsley Biomedical Research Centre, South London and Maudsley NHS Trust, London, UK. ✉e-mail: p.a.demange@vu.nl; rosa.cheesman@kcl.ac.uk

education's primary focus: academic and cognitive performance. The umbrella of non-cognitive skills encompasses a wide range of competencies, including academic motivation, perseverance, mind-sets, learning strategies, and social skills[4,5]. Cognitive skills like executive functioning, working memory, and verbal IQ are more precisely integral to cognitive functioning, but both cognitive and non-cognitive skills are critical for educational success[4]. Research in developmental psychology[6], economics[7], and sociology[8] has suggested that parents socially influence their children's non-cognitive skills including emotion regulation, social capacities, attitudes and motivations[9,10]. Given that non-cognitive skills (particularly self-control and emotion regulation[11,12]) support education, it follows that parents' non-cognitive skills may also affect children's educational outcomes.

Prior research has detected small associations between measured parental non-cognitive skills and offspring educational outcomes. In one study, mothers' locus of control was the only significant non-cognitive predictor of offspring college attendance ($\beta = 0.02$, $p < 0.05$; $\beta = -0.01$ for maternal self-concept and self-esteem, both non-significant)[13]. Mothers' cognitive skills, measured by the U.S. Armed Forces Qualifying Test, were a stronger predictor ($\beta = 0.06$, $p < 0.01$). Another study found that fathers' non-cognitive skills were associated with sons' standardised test scores at age 16 ($\beta = 0.09$)[14]. Here, non-cognitive skills were measured by a single composite of extraversion, neuroticism, persistence, and perseverance from a standardised Swedish military-oriented psychological evaluation. Additionally, parents' attitudes towards education and social skills have been found to account for 8% of the socioeconomic gap in children's achievement[15]. The contributions of specific measured parental traits that were included were also not stated.

Two key limitations weaken this base of evidence on the effects of parents' skills on offspring education: challenges with phenotypic assessments of parents' non-cognitive skills, and genetic confounding.

First, regarding assessment, whereas cognitive skills can be directly measured by tests of domain-specific or general cognitive performance, non-cognitive skills are more challenging to capture[16,17]. There is little agreement on what non-cognitive skills to measure. Some researchers focus on personality, whereas others include self-control, self-esteem, motivation, and interests. Importantly, studies identifying partial effects of specific parental cognitive and non-cognitive skills are less informative about the overall influences of these domains. Measurement error could also mean that effects of parents' non-cognitive skills have been underestimated.

Genetic methods offer an alternative approach to defining parents' non-cognitive skills. Both cognitive and typically-studied non-cognitive skills are substantially genetically influenced, with twin study heritability estimates of 40–70%[18,19]. A new method–'GWAS-by-subtraction'–makes it possible to assess a broad latent genetic non-cognitive construct, by 'subtracting' cognitive ability-related genetic variation from educational attainment genetic variation[20]. This follows an influential definition of non-cognitive skills from economics[21] as all traits positively contributing to educational and professional success that are not cognitive skills. This non-cognitive genetic construct–which could otherwise be conceptualized as 'not-cognitive'–is associated with higher socioeconomic attainment, more open and conscientious personality, and some psychiatric disorders (e.g., higher risk for schizophrenia, lower risk for attention deficit/hyperactivity disorder). In the present study, we use this GWAS-by-subtraction measure of non-cognitive skills to capture the overall effect of all non-cognitive parent phenotypes on offspring education. This phenotype-agnostic approach is somewhat loose: it could include parental phenotypes not traditionally classed as 'non-cognitive' or 'skills'. However, it provides a useful first step towards characterizing pathways from specific parental skills to offspring educational outcomes. After establishing overall effects, complementary research

designs using measured parental non-cognitive skills can subsequently be used to identify specific mediating mechanisms.

A second challenge is to distinguish social (i.e., environmental) from genetic transmission. None of the associations between parental skills and offspring education cited above were estimated using genetically sensitive designs. This is problematic, because from just parent-offspring correlations one cannot conclude that parents' skills shape offspring education, for instance by providing resources, experiences, and support. Ignoring any shared genetic influences on parents' skills and child educational outcomes confounds estimation of the effects of parental phenotypes on offspring outcome[22]. To establish the extent that parents' (non-)cognitive skills influence child educational outcomes socially, it is vital to control for inherited genetic effects.

Genetic study designs can isolate environmental effects of parental skills on offspring education, controlling for genetic transmission. Several designs estimate a genetic effect of the child's genotype on the child phenotype (direct genetic effect), and an environmentally mediated effect of the parental genotype on the child's phenotype (parental indirect genetic effect). For example, non-transmitted genetic variants affect offspring phenotypes indirectly via the environment shaped by parental phenotypes[23,24]. Polygenic scores (individual-level indices of trait-specific genetic endowment; PGS) for educational attainment based on parents' non-transmitted variants, are associated with offspring attainment[25–27]. Complementary evidence of indirect effects of parents' education-linked genetics on offspring education has also accumulated from sibling and adoption PGS designs[25,26,28,29]. To obtain estimates of indirect genetic effects using sibling data, within-sibling genetic associations (first developed to estimate direct genetic effects independent of population biases[30,31]) are compared to population-based associations. To obtain estimates of indirect genetic effects using adoption data, genetic associations estimated for adoptees and non-adopted individuals are compared[29]. Notably, variance decomposition as well as PGS methods can be applied to disentangle direct and indirect genetic effects, but the former requires much larger sample sizes[32–35]. It is not known whether parental indirect genetic effects on offspring education occur through cognitive or non-cognitive pathways (or both), because studies have not parsed out the contributions of subcomponents of the educational attainment PGS.

Here, we directly compare estimates of parental indirect genetic effects obtained from different designs. Estimation of genetic associations may involve numerous biases[36–38]. Sibling, adoption, and non-transmitted allele designs have different assumptions and subtle differences in biases and components affecting the estimated indirect genetic effect. As shown by our data simulations indirect genetic effect estimates from the adoption design may be less biased by population stratification and assortative mating than the sibling and non-transmitted allele designs (see Supplementary Note 6 and our GitHub repository[39]). However, estimates obtained from the adoption design do not capture prenatal parental environmental effects on child education and may be less generalisable to the population. The sibling design may estimate parental indirect genetic effects with more bias from sibling genetic effects. Triangulation across designs and sensitivity analyses can help detect possible biases and quantify parental indirect genetic effects and other environmental effects[37,40].

In the current study (pre-registration: https://osf.io/mk938/), we use a novel approach to estimate the social effects of parents' cognitive and non-cognitive skills on offspring education. We deploy GWAS-by-subtraction to estimate individuals' genetic endowments (PGS) for cognitive and non-cognitive skills, and test how much these operate environmentally via parental influences on offspring educational outcomes. We provide a comparison of parental indirect genetic effects in three cohorts of genotyped families in two countries (UK Biobank, UK Twins Early Development Study, Netherlands Twin

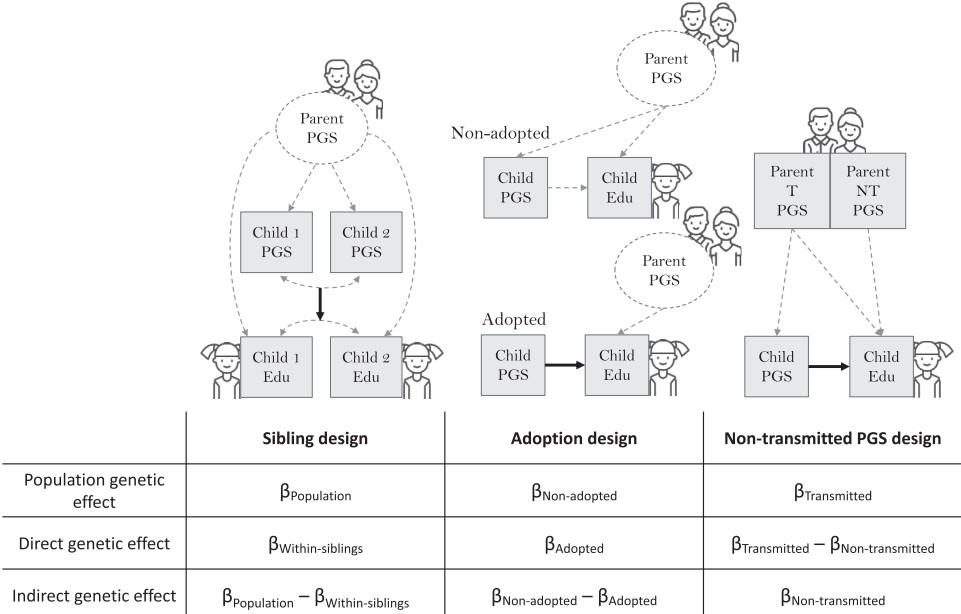

**Fig. 1 | Analytical designs to estimate direct and parental indirect genetic effects.** Square = observed variable, circle = unobserved/latent variable; β = estimated effect of polygenic score (PGS) on outcome; the population effect of a PGS captures both direct and indirect genetic effects; direct genetic effects (controlling for indirect genetic effects) are represented with solid arrows. Icons made by Freepik from www.flaticon.com.

Register). Each cohort includes multiple achievement outcome measures (i.e., standardised test results and teacher-reported grades in childhood and adolescence) and attainment (i.e., years of completed education reported in adulthood). We triangulate across three complementary study designs for estimating parental indirect genetic effects and assess the presence of components and biases.

## Results

### GWAS-by-subtraction results
We identified the genetic components of cognitive and non-cognitive skills using Genomic SEM, following Demange et al.[20], in samples that excluded participants used for polygenic score analyses. Educational attainment and cognitive performance meta-analytic summary statistics (see Methods) were regressed on two independent latent variables, Cog and NonCog (see Supplementary Fig. 1). These two latent factors were then regressed on 1,071,804 HapMap3 SNPs in a genome wide association (GWA) design. The LD score regression-based SNP heritabilities of Cog and NonCog were 0.184 (SE = 0.007) and 0.054 (SE = 0.002), respectively. More information on the GWAS is presented in Supplementary Data 1.

### Descriptive statistics
SNP associations with the Cog and NonCog latent variables provided the weights to create individual-level polygenic scores in 3 cohorts with family data and educational achievement and/or attainment outcomes. Sample sizes for individuals with polygenic score and educational outcome data were: 39,500 UK Biobank siblings, 6409 UK Biobank adoptees, up to 4796 DZ twins in the Twins Early Development Study (TEDS), up to 3163 twins and siblings in the Netherlands Twin Register (NTR), and up to 2534 NTR individuals with both parents genotyped. Full phenotypic descriptive statistics are available in Supplementary Data 2.

### Overview of three family-based polygenic score designs
To estimate direct offspring-led and indirect parent-led effects of PGS for cognitive and non-cognitive skills on educational outcomes, we considered three family-based genomic designs. The designs are

illustrated in Fig. 1. All models jointly included Cog and NonCog PGS. Note that population effects are equivalent to PGS effects estimated in standard population analyses that do not use within-family data. In contrast, within-family designs exploit the principles of Mendelian segregation or the natural experiment of adoption to separate direct and indirect/social components of the overall population PGS effect. Importantly, a direct genetic effect is only direct in the sense that it does not originate from another individual's genotype. Direct effects are also not 'purely' genetic, but lead to educational outcomes via intermediate pathways, and are expressed in the context of environments.

First, the sibling design estimates indirect genetic effects by comparing population-level and within-family (i.e., within-sibling or within-DZ twin) PGS associations (Eq. (1))[28]. The direct effect of a polygenic score is estimated based on genetic differences between siblings, which are due to random segregations of parental genetic material, independent of shared family effects (including parental indirect genetic effects). Specifically, the direct effect is estimated using a variable representing individuals' polygenic scores minus the average polygenic score for their family: the within-family beta (β_Within in Eq. (1)). The population effect of a polygenic score is estimated in a separate model, simply regressing the outcome variable on polygenic score differences between individuals from different families (Eq. (2)). The indirect genetic effect is obtained by subtracting the within-family PGS effect estimate from the population effect estimate.

$$EA_{ij} = \alpha_0 O + \beta_{\text{Within}_{Cog}}\left(PGS_{Cog_{ij}} - \overline{PGS_{Cog_j}}\right)$$
$$+ \beta_{\text{Between}_{Cog}}(\overline{PGS_{Cog_j}}) + \beta_{\text{Within}_{NonCog}}(PGS_{NonCog_{ij}} - \overline{PGS_{NonCog_j}}) \quad (1)$$
$$+ \beta_{\text{Between}_{NonCog}}(\overline{PGS_{NonCog_j}}) + Z_{ij}$$

$$EA_{ij} = \alpha_0 O + \beta_{Cog}(PGS_{Cog_{ij}}) + \beta_{NonCog}(PGS_{NonCog_{ij}}) + Z_{ij} \quad (2)$$

Note: EA is the educational outcome, PGS is the polygenic score (for Cog PGS_Cog and NonCog PGS_NonCog). $\overline{PGS}$ refers to the average polygenic score in the family j. i refers to the individual sibling. $\alpha_0$ refers to the intercept, Z are covariates for the individual i: sex, age, sex*age, the first 10 principal components, and genotyping platform.

See Supplementary Note 5 for a comparison of different versions of this sibling design, using data simulations.

Second, indirect genetic effects can be estimated by comparing polygenic score associations estimated in a sample of adoptees against those estimated for individuals who were reared by their biological parents[29]. Therefore, we estimate the regression model shown in Eq. (2) separately for adoptees and for non-adopted individuals. The population effect is estimated as the polygenic score effect on phenotypic variation among non-adopted individuals (i.e., a combination of direct and indirect genetic mechanisms). The direct genetic effect is the effect of the polygenic score among adoptees. Adoptees do not share genes by descent with their adoptive parents, so we expect their polygenic scores to be uncorrelated with the genotypes of their adoptive parents. Therefore, the polygenic score effect in adoptees cannot be inflated by environmentally mediated parental indirect genetic effects. In this design, the indirect genetic effect is estimated by subtracting this direct PGS effect from the population effect estimated in the non-adopted group. When taking the difference, it is important that the groups are similar in terms of all observed and unobserved confounders, an untestable assumption that is unlikely to always hold. We found small differences between adoptees and non-adopted individuals in the UK Biobank in their demographic and early-life characteristics. Cohen's d values were $d < 0.15$ for Cog and NonCog PGS and educational attainment, and $d = 0.31$ for birth weight. The pattern of geographical clustering of adopted and non-adopted participants across the UK was highly similar (see Supplementary Data 11, Supplementary Note 3, and Supplementary Fig. 2).

Third, indirect genetic effects can be estimated, and disentangled from direct genetic effects, using information on parental genetic variation that was not transmitted to offspring[25,26] (Eq. (3)).

$$EA = \alpha_0 0 + \beta_{T_{Cog}}(PGS_{T_{Cog}}) + \beta_{TNonCog}(PGS_{T_{NonCog}}) \\ + \beta_{NT_{Cog}}(PGS_{NT_{Cog}}) + \beta_{NT_{NonCog}}(PGS_{NT_{NonCog}}) + Z \quad (3)$$

The population effect is estimated from a polygenic score based on transmitted variants ($\beta_T$). Transmitted genetic variants are present in an offspring and in at least one of their parents, and so may influence offspring education via both direct and indirect mechanisms. The parental indirect genetic effect is estimated as the effect of a polygenic score based on parental variants that were not transmitted to offspring ($\beta_{NT}$). Non-transmitted variants can only take effect on offspring education through the environment. The direct genetic effect is estimated by partialling out the effect of the non-transmitted polygenic score from that of the transmitted polygenic score ($\beta_T - \beta_{NT}$). Maternal and paternal scores are averaged to create overall parental non-transmitted polygenic scores. We did not distinguish between maternal and paternal PGS, due to the replicated evidence that mothers' and fathers' PGS for educational attainment have equal effects on offspring education[41,42], and to enable closer comparison with the adoption and sibling designs, which yield estimates of the overall parental genetic effect. Notably, regressing offspring phenotype on offspring PGS and parental PGS would allow equivalent estimation of the parental indirect genetic effect without haplotype estimation[43].

### Parents' heritable cognitive and non-cognitive skills environmentally influence offspring education

We investigated environmental effects of parents' non-cognitive and cognitive skills on offspring education by estimating parental indirect genetic effects of NonCog and Cog PGS. Figure 2a shows that, for both NonCog and Cog PGS, indirect genetic effects of parents on offspring education were present (meta-analytic indirect $\beta_{NonCog} = 0.08$, SE = 0.03; indirect $\beta_{Cog} = 0.10$, SE = 0.01), in addition to direct genetic effects (direct $\beta_{NonCog} = 0.14$, SE = 0.03; direct $\beta_{Cog} = 0.15$, SE = 0.02). Averaged across all designs, outcomes and cohorts, indirect

environmentally mediated effects explained 36% of the population effect of the NonCog PGS, and 40% of the population effect of the Cog PGS. However, results varied depending on the methods used and outcomes investigated. Results per cohort, outcome and design, as well as population genetic effects and the ratio of indirect to population effects are reported in Supplementary Data 3 and Supplementary Figs. 3, 4 and 5. Meta-analytic results are reported in Supplementary Data 4. Z-tests results comparing direct and indirect effects are reported in Supplementary Data 5.

### Estimates of indirect genetic effects vary by age, outcome and cohort

Figure 2b shows estimates of direct and indirect genetic effects of NonCog and Cog PGS for different cohorts and educational outcomes, holding the design constant (i.e., the sibling design, which was available for all cohorts and outcomes). Estimates were highly consistent across cohorts except for age 12 achievement in Dutch versus UK cohorts: indirect genetic effects were negligible and represented a small fraction of the population effect in NTR (3% and 23% for NonCog and Cog, respectively), whereas they accounted for 56% and 48% of the population effects of NonCog and Cog PGS in TEDS. For adult educational attainment, estimates of direct and indirect effects were more similar for the Dutch (NTR: indirect $\beta_{NonCog} = 0.11$, SE = 0.03; indirect $\beta_{Cog} = 0.06$, SE = 0.03) and UK (UKB: indirect $\beta_{NonCog} = 0.12$, SE = 0.01; indirect $\beta_{Cog} = 0.12$, SE = 0.01) cohorts. See Supplementary Data 3 for full results.

### Estimates of indirect genetic effects depend on the analytical design

Figure 2c shows estimates of direct and indirect genetic effects of NonCog and Cog PGS for different designs, holding the phenotype constant (i.e., educational attainment, which was available for all three methods). While estimates obtained with sibling and non-transmitted PGS methods indicate equal indirect effect sizes (indirect βs for educational attainment ranged between 0.11 and 0.12; see Supplementary Data 3 and 4), the adoption design yielded low to null indirect genetic effects for both NonCog and Cog PGS (indirect $\beta_{NonCog} = 0.02$, SE = 0.02; indirect $\beta_{Cog} = 0.08$, SE = 0.02).

Figure 3 summarises how the three designs estimate parental indirect genetic effects in the presence of different contributors, thus highlighting possible explanations for lower adoption-based estimates. This information is based on simulations (see Supplementary Notes 4 and 6, Supplementary Fig. 9, and our GitHub repository[39]). We consider prenatal and postnatal parental indirect genetic effects as components of the total parental indirect genetic effect, and other simulated contributors as biases. First, unlike the sibling and non-transmitted allele designs, the adoption design does not capture indirect genetic effects occurring in the prenatal period. Second, the adoption design estimates indirect genetic effects with less bias from population stratification. Third, the adoption design estimates indirect genetic effects with less bias from assortative mating than the sibling design, and, most likely, than the non-transmitted alleles design. How the bias in the adoption design estimates compares to the non-transmitted design depends on the precision of the polygenic score, see Supplementary Note 6. Any excess indirect genetic effect estimated in the sibling/non-transmitted allele designs compared to the adoption design therefore indicates the overall impact of prenatal indirect genetic effects, population stratification, and assortative mating. Sibling indirect genetic effects are an important potential influence to consider, but cannot explain the empirical results because they only do not affect indirect effect estimates from adoption and non-transmitted designs differently (they mainly inflate sibling-based estimates).

With the adoption design, the indirect genetic effect of the NonCog PGS on educational attainment in UK Biobank is 83% lower than with the sibling design, while it is only 33% lower for Cog. This

## a. Meta-analytic results

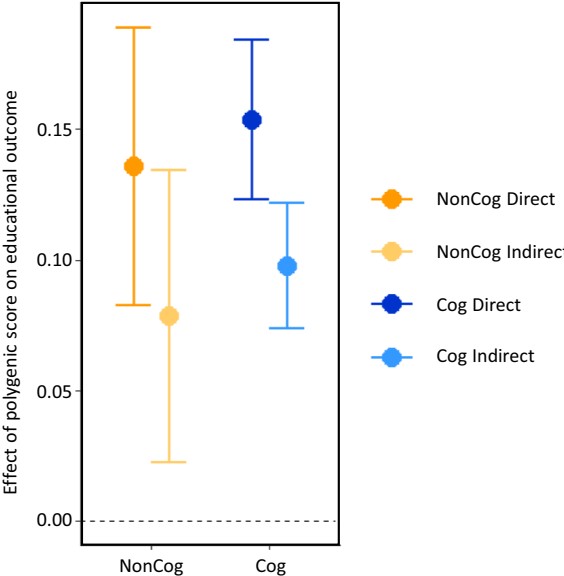

## b. Sibling design by cohort

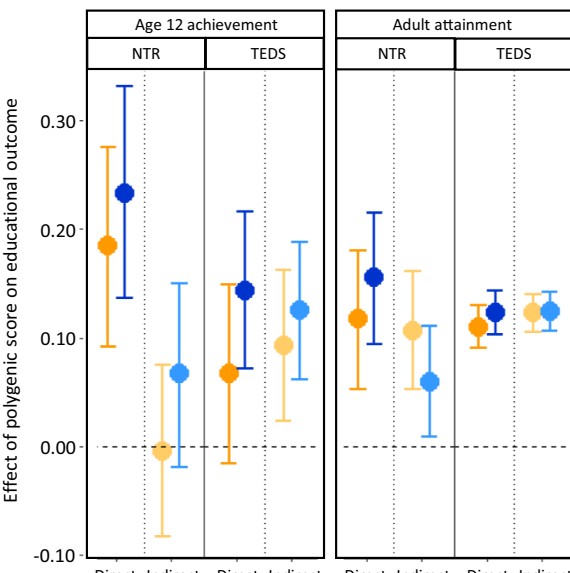

## c. Educational attainment by design

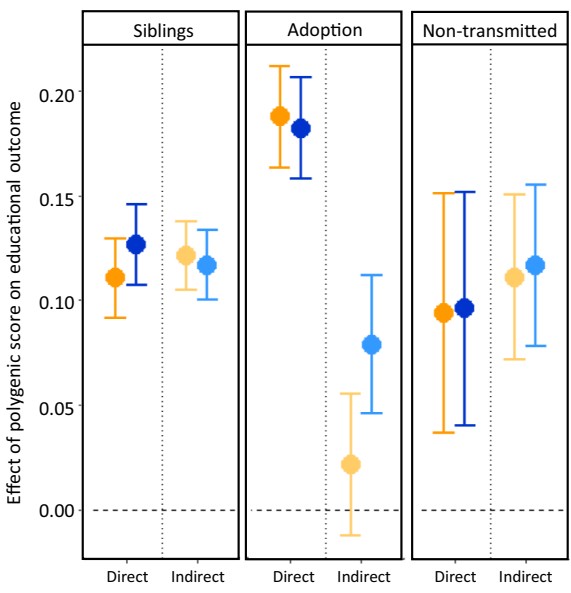

**Fig. 2 | Estimated direct and indirect genetic effects of NonCog and Cog PGS on educational outcomes. a** Meta-analytic results. Meta-analysed estimates of direct and indirect genetic effects of NonCog and Cog PGS on educational outcomes (N = 68,919). Indirect genetic effects work through the environment that parents provide for their children. Beta coefficients were obtained from meta-analysis of effects across cohorts, designs and outcome phenotypes; bars = 95% CIs. **b** Sibling design by cohort. Estimates of direct and indirect effects of NonCog and Cog PGS by cohort (for age 12 and adult outcomes), using the sibling design only. NTR is a Dutch cohort (N = 1631 and N = 3163 respectively), TEDS (N = 2862) and UKB (N = 16,624) are UK cohorts; bars = 95% CIs. **c** Educational attainment by design. Estimates of direct and indirect effect of NonCog and Cog PGS by analytic design (for adult educational attainment outcomes only). Samples sizes: N = 42,663 (results meta-analysed across UKB and NTR); N = 6407 adoptees and 6500 non-adopted individuals (UKB); N = 2534 trios in NTR; bars = 95%CIs.

suggests that estimates for NonCog are affected more strongly than Cog by population stratification, assortative mating and/or prenatal indirect genetic effects.

### Population phenomena may inflate indirect genetic effect estimates

Although triangulating designs suggested that prenatal indirect genetic effects, population stratification, and assortative mating may contribute to the higher estimated parental indirect genetic effects from non-transmitted alleles/sibling designs relative to the adoption design, this approach cannot disentangle the relative importance of these individual biases. To this end, we conducted additional sensitivity analyses to assess the magnitudes of these biases (not pre-registered).

First, we analysed the GWAS summary data on which the polygenic scores were based, using LD score regression to detect population stratification. The LD score regression ratio statistics of uncorrected educational attainment and cognitive performance GWAS were 0.11 (SE = 0.01) and 0.06 (SE = 0.01), respectively (Supplementary Data 1). These non-null estimates indicated that a small but significant portion of the GWAS signal was potentially attributable to residual population stratification. As cognitive performance seems

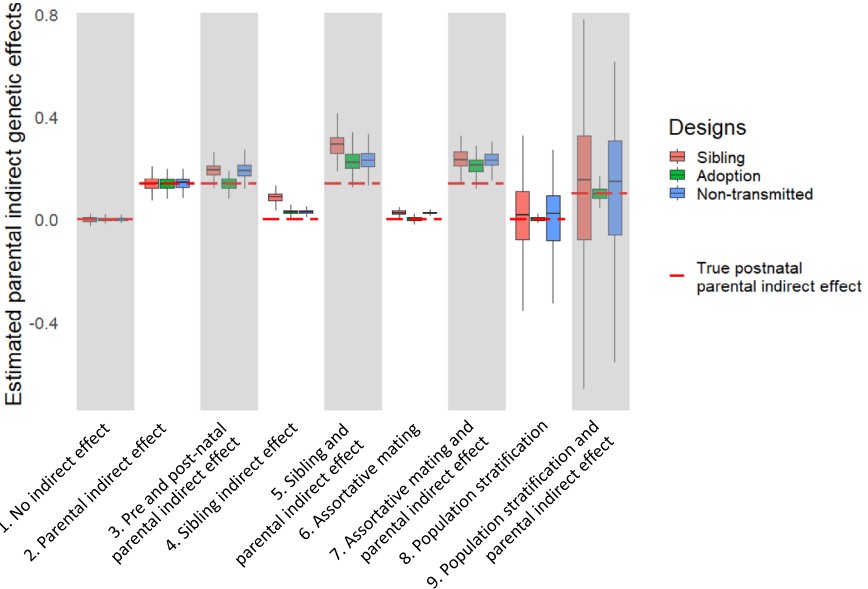

**Fig. 3 | Estimates of parental indirect genetic effects from the three designs, based on data simulated to include different components and biases.** Components include parental prenatal and postnatal indirect genetic effects. Biases include sibling indirect genetic effects, assortative mating, and population stratification. Boxplots of 100 replicates based on a simulated sample of 20,000 families. The center line represents the median, the box limits are the 1st and 3rd quartile, and the whiskers reach the maximum value within 1.5 times the interquartile range. Outlying values are not represented. For clarity, the red line benchmarks the true simulated *postnatal* parental indirect effect, although we note that *prenatal* parental genetic effects are a component rather than a bias of the parental indirect genetic effect.

less prone to population stratification than EA, it is possible our estimates of direct and indirect genetic effects of NonCog were more biased by population stratification than Cog.

Second, we detected slight evidence of assortative mating, which appeared stronger in the UK than Dutch cohorts. In NTR, parental PGS correlations are non-significant (NonCog $r = 0.03$, Cog $r = 0.02$). Sibling PGS intraclass correlations ranged between 0.49 and 0.52 in NTR, and between 0.53 and 0.56 in TEDS and UK Biobank. This supports the presence of assortative mating on NonCog and Cog PGS potentially biasing our estimates of indirect genetic effects in UK cohorts, but less in our Dutch cohort. See Supplementary Data 6 for full correlations.

Third, our sensitivity analyses did not support the presence of indirect effects of siblings' NonCog and Cog PGS on individuals' educational outcomes. Our first approach leveraged sibling polygenic scores, the rationale being that in the presence of a sibling effect, a sibling's PGS will influence a child's outcome beyond child and parent PGS. In NTR, siblings' NonCog or Cog PGS had non-significant effects (Supplementary Data 7). In a second approach, the difference in PGS effects on EA between monozygotic (MZ) and dizygotic (DZ) individuals was tested. Since MZ twins are more genetically similar than DZ twins, their PGS should capture more of the indirect genetic effect of their twin. In NTR and TEDS, PGS effects were not significantly different between MZs and DZs (Supplementary Data 8 and Supplementary Fig. 6). Finally, in UKB, we tested PGS effects on EA given the number of siblings individuals reported having. If more siblings lead to a stronger sibling effect, this will be captured as an increased effect of an individual's own PGS on the outcome in the presence of more genetically related siblings. As a negative control, we conducted the same analysis in adoptees. Since adoptees are unrelated to their siblings, their PGS do not capture sibling effects at any family size. NonCog PGS effects weakly increased with number of siblings, but this pattern was also present in adoptees, suggesting confounding by unobserved characteristics of families with numerous children (Supplementary Data 9 and Supplementary Fig. 7).

## Discussion

We used genetic methods to study environmental effects of parents' skills on child education. We found evidence that characteristics tagged by NonCog and Cog polygenic scores (PGS) are both involved in how parents provide environments conducive to offspring education. Indeed, indirect genetic mechanisms explained 36% of the population effect of the NonCog PGS, and 40% of the population effect of the Cog PGS (population $\beta_{NonCog} = 0.22$, SE = 0.01; population $\beta_{Cog} = 0.25$, SE = 0.01). This result was consistent across countries, generations, outcomes, and analytic designs, with two notable exceptions. First, estimated parental indirect genetic effects were null for childhood achievement in our Dutch cohort (NTR), but not for comparable outcomes in our UK cohort (TEDS). Second, parental indirect genetic effects estimated with the adoption design were lower than for the sibling and non-transmitted allele designs, particularly for the NonCog PGS. Given our evidence from data simulations that the adoption-based estimates of indirect genetic effects do not account for prenatal effects and may be more robust to population stratification and assortative mating, these factors may contribute substantially to estimates from the other two designs, especially for the NonCog PGS. This was supported by results from sensitivity analyses.

This study demonstrates the utility of genetic methods for assessing elusive phenomena: non-cognitive skills, and genuine environmental influences from parents, unconfounded by offspring-led effects of inherited genes. Compared to analysing a set of measured parental non-cognitive skills, our GWAS-by-subtraction approach captures a wider array of traits linked genetically to attainment, and therefore broadly quantifies the overall salience of parents' non-cognitive skills. Our evidence that parents' non-cognitive and cognitive skills are both important for children's education complements the growing literature that has considered effects of specific measured skills within both of these domains[13,14]. These studies found that effects of parents' non-cognitive skills on offspring education were less than half the size of the effects of

parents' cognitive skills. In contrast, we found that indirect genetic effects of NonCog PGS were almost as large as for Cog skills. This discrepancy might stem in part from our comprehensive definition of non-cognitive skills, as we do not rely on possibly unreliable and incomplete phenotypic measures. Importantly, the parental indirect genetic effects we have identified may capture proximal forms of 'nurture' (e.g., a parent directly training their child's cognitive skills, or cultivating their child's learning habits through participation and support) and/or more distal environmental effects (e.g., a parent's openness to experience leading them to move to an area with good schools). The environmental effects of parents' non-cognitive and cognitive skills are likely to be larger than we estimate, because our approach only captures effects of parent skills tagged by current GWAS. Polygenic scores index a subset of the common genetic component of parent skills, which is in turn a fraction of the total genetic component (missing heritability[44,45]), and cannot account for the non-heritable component of parent skills.

The lower importance of parental indirect genetic effects for child achievement in the Netherlands compared to the UK indicates that our UK achievement outcomes more strongly capture variation in family background. This difference could result from the design of these achievement measures: Dutch test results are standardized based on a representative population, but UK teacher reports might still be affected by student social background. Societal differences between the two countries might offer another explanation, as indirect genetic effects might be seen as indicator of social inequality (similarly to shared-environment variance in twin studies[46]). For adult attainment, results were more similar across UK and Dutch cohorts, corresponding with recent evidence for consistent shared-environment influence on educational attainment across social models[47]. This consistency also suggests that the difference in childhood is not due to a cohort or population difference. The higher indirect genetic effects in adult attainment in the Netherlands might reflect an increase in environmental variance following tracking taking place in secondary schools[27]. Indeed, socioeconomic disparities in achievement seem to increase more between ages 10 and 15 in the Netherlands than in the UK[48] and children whose parents have a higher education are more likely to enrol in a higher educational track independently of their achievement at age 12[49], suggestive of greater parental effects on secondary and later education, which should be tested in further studies.

We found that the design used to estimate indirect genetic effects matters, with the adoption design giving systematically lower estimates. Direct comparison of results across designs suggested that 33% (for Cog) and 83% (for NonCog) of the indirect genetic effects on adult educational attainment, estimated using the sibling design, are at least in part due to population stratification, assortative mating, and prenatal indirect genetic effects. The importance of population stratification for genetic associations with educational attainment was suggested by recent UK Biobank studies[50,51]. Our sensitivity analyses also indicated residual population stratification, which was more severe for the NonCog GWAS. There was some evidence of assortative mating, with sibling PGS correlations above expectation (>0.5) particularly in the UK cohorts. This country difference in assortment is supported by the lower estimated spouse PGS correlations in NTR (0.02 for Cog, 0.03 for NonCog) than for the EA PGS in the UK Biobank (0.06)[52]. There was no statistically significant difference in assortative mating between Cog and NonCog, suggesting that population stratification explains the particularly large design-based discrepancy between estimates of indirect genetic effects for NonCog (but possibly also differential bias in the Cog versus NonCog GWAS; see Limitations). Population stratification should be carefully considered in studies using NonCog PGS. Structural equation models, leveraging within-family polygenic scores and phenotypes, are being developed to parse the

contributions of indirect and direct genetic effects to complex traits from assortative mating (both disequilibrium and equilibrium forms) and population stratification[53,54]. Another consideration for future research is that indirect genetic effects on education might span across more than a single generation, for example the influence of grandparents. Since cumulative indirect genetic effects are all removed when a child is adopted, their presence would contribute to the observed difference in indirect effect between the adoption and other designs.

Regarding siblings, we did not find evidence that indirect effects of siblings' NonCog and Cog PGS affect individual differences in educational outcomes, using three different approaches. This corresponds with null findings regarding indirect effects of siblings' educational attainment genetics in the UK Biobank[50,51]. However, other UK Biobank studies have detected indirect effects of older siblings' EA PGS on younger siblings' educational attainment[55], and parental compensation for sibling EA PGS differences[56], suggesting that more subtle analyses are required to understand sibling effects. There is also some evidence for sibling effects on educational attainment in other populations, based on the EA PGS[26] and on extended twin family data[57]. It is possible that our PGS analyses were not sufficiently powered to detect indirect genetic effects of siblings, since they were based on lower sample size than our main analyses. However, our results suggest that indirect genetic effects of siblings on education are small. Therefore, our methods provide good proxies for parental indirect genetic effects, with minimal inflation from sibling effects.

Our data suggest that the adoption design may provide a useful lower-bound estimate of indirect genetic effects of parents. Given that there was no evidence for sibling effects of the Cog or NonCog PGS, our adoption-based estimates, which appear to be less biased by population stratification and assortative mating, should give a closer measure of (postnatal) parental indirect genetic effects in the absence of other issues. However, adoptees and non-adopted individuals differ in unobserved and observed ways, including birthweight ($d = 0.3$). These differences likely make adoption-based estimates of indirect genetic effects, which rely on a comparison of the two groups, less reliable. Moreover, three additional factors may make the adoption-based estimates of indirect genetic effects too conservative. First, adoption based indirect effect estimates exclude prenatal indirect genetic effects (and indirect genetic effects taking place between the birth and moment of adoption), which might influence educational outcomes[58,59]. While we are unable to test for prenatal indirect effects, these could be investigated in cohorts with pregnancy information, adjusting for postnatal indirect genetic effects. Second, adoptees may have been exposed to a narrower range of environments (e.g., family socioeconomic status) compared to non-adopted individuals[60]. This form of selection bias is likely to increase the genetic variance at the expense of the indirect genetic effect. Third, selective placement of children in adoptive families matching characteristics of their biological families, or adoption of children by close relatives[61], could result in correlation between child and (adoptive) parent genotypes, leading to an underestimation of the indirect genetic effect. There is modest evidence for selective placement of adoptees based on education in the US[62]. We cannot control for selection and relatedness (e.g., by excluding individuals who were adopted by relatives and/or adopted relatively late in development), since there is no information on the adoptive parents in the UK Biobank resource.

We acknowledge several limitations. First, while we suggest that an attribute of our study is the broad and phenotype-agnostic characterisation of non-cognitive skills, our GWAS-by-subtraction approach is unable to identify specific parental characteristics and is also still limited by measures of cognitive performance and educational attainment in the original GWAS. Some cognitive skills might

not be reflected in the available Cognitive Performance GWAS, so the NonCog factor could capture genetic influences affecting cognition. However, previous analyses have shown that a NonCog PGS based on GWAS-by-subtraction predicts substantially less variation in cognition than the Cog PGS[20]. Additionally, our NonCog latent variable reflects the residual variance of adult educational attainment, and therefore is a measure of non-cognitive aspects of adult EA. Non-cognitive aspects of childhood achievement might differ somewhat, which might lead to an underestimation of indirect genetic effects of the NonCog PGS on these outcomes.

Second, the generalisability of our results is limited. Highly educated individuals are over-represented in all cohorts. Participation bias also affects GWAS results[63]. Selection effects may be especially strong in the adoption design as adoptions may depend on (partially heritable) phenotypes of the biological parents, and many adoptive parents are also selected based on their (partially heritable) behavioural phenotypes. Additionally, only participants of European descent were included in the analysis.

Third, replication efforts are needed. Special effort should be targeted to include diverse ancestry participants. While our overall estimates are well powered due to the aggregation of cohorts, some analyses rely on a single sample. As such, results from these analyses might reflect specifics of these samples and not design-specific biases and should be replicated.

Fourth, although our within-family methods allowed us to evaluate biases in polygenic score effects within the target samples, the same biases are likely to influence the effect size estimates from the original population-based GWAS used to construct polygenic scores. This problem has been explored in relation to the sibling design in a recent preprint[64], but remains to be investigated for non-transmitted PGS and adoption designs. Population GWAS effects could be differentially affected (i.e., stronger correlation between direct and indirect genetic effects) for NonCog versus Cog, which would make their respective PGS effects less comparable. Increasingly large within-family GWAS[35,65] of Cog and EA will allow this to be resolved.

Finally, while we conceptualize our NonCog PGS as a non-cognitive measure, it could also be considered a 'not-cognitive PGS', since it is a residual construct that results from removing heritable variance associated with cognitive skills from the heritable variance in educational attainment. In the future, it may be useful to develop a more precise non-cognitive skills GWAS, by creating the latent Cog and NonCog factors using additional measured phenotypes. To this end, large GWA meta-analyses should be completed not only for personality[66] but for other classic non-cognitive skills such as motivation and self-control.

Several additional future research directions emerge. First, given that we have quantified the overall environmental effects of parents on offspring education tagged by NonCog and Cog PGS, the next step is to identify specific mediating parent characteristics, whether proximal or distal. It will be informative to test not only typical non-cognitive skills measures such as parental locus of control (as suggested by[13]), but also 'not-cognitive' characteristics that do not appear in non-cognitive skill batteries yet are genetically correlated with the NonCog PGS and phenotypically correlated with offspring achievement. For instance, parental depression is a feasible partial mediator, given that Major Depressive Disorder is significantly genetically correlated with NonCog ($r_g = -0.19$, $p = 2.62E{-}14$)[20], and maternal depression is associated with offspring mathematics performance, possibly via offspring executive function[67]. Researchers could also examine mediating child characteristics on the pathway between their parents' characteristics and their own educational outcomes. Children's skills themselves might not be involved in these pathways. Indeed, educated parents do not appear to affect offspring education by fostering non-cognitive skill development[11], and twin

research shows no influence of shared environmental factors on individual differences in children's measured non-cognitive skills such as grit and self-control[68–70].

A second future direction is to incorporate gender and socioeconomic status into research on indirect genetic effects on education. Twin data show that shared environmental contributions to educational attainment are larger for women than for men[47]. It is unknown whether this finding holds for indirect genetic effects and for childhood achievement. Another gender aspect to consider is differential maternal and paternal indirect genetic effects[33]. There is some evidence (although not genetically informed) that mother and father skills show unique associations with offspring education[14]. Indirect effects of parents' genetic endowment for non-cognitive skills on child education might be mediated or moderated by parents' income and cultural capital (including school-related skills and habits). While some evidence suggests that home learning environments may be more cognitively stimulating in families of higher socioeconomic[71,72], there is also evidence suggesting that mothers who have lower reported incomes also report more frequent activities that facilitate cognitive stimulation[73].

In sum, this study provides evidence for environmental effects of parents' non-cognitive and cognitive skills on offspring educational outcomes, indexed by indirect genetic effects of polygenic scores. Combining three cohorts and three designs for estimating indirect genetic effects allowed us to obtain robust findings. These results have significance for human health, as the role parents play in successful cognitive development and (mental) health development go hand in hand.

## Methods

Our research complies with all relevant ethical regulations. Project approval for the Twins Early Development Study (TEDS) was granted by King's College London's ethics committee for the Institute of Psychiatry, Psychology and Neuroscience PNM/09/10–104. Ethical approval for the Netherlands Twin Register (NTR) was provided by the Central Ethics Committee on Research Involving Human Subjects of the VU University Medical Center, Amsterdam, and Institutional Review Board certified by the U.S. Office of Human Research Protections (IRB number IRB-2991 under Federal-wide Assurance-3703; IRB/institute codes 94/105, 96/205, 99/068, 2003/182, 2010/359) and participants provided informed consent. The UK Biobank has received ethical approval from the National Health Service North West Centre for Research Ethics Committee (reference: 11/NW/0382). Informed consent was obtained from all human participants.

The study methods were pre-registered on the Open Science Framework (https://osf.io/mk938/) on the 24/02/2020. Additional non-preregistered analyses are indicated as such below and should be considered exploratory. Additional deviations from the pre-registration are detailed in Supplementary Note 1.

### Samples
**UK Biobank.** The UK Biobank is an epidemiological resource including British individuals aged 40 to 70 at recruitment[74]. Genome-wide genetic data came from the full release of the UK Biobank data, and were collected and processed according to the quality control pipeline[75].

We defined three subsamples of the UK Biobank to be used for polygenic score analyses: adopted participants, a control group of non-adopted participants, and siblings. Starting with UK Biobank participants with QC genotype data and educational attainment data ($N = 451{,}229$), we first identified 6407 unrelated adopted individuals who said yes to the question "Were you adopted as a child?" (Data-Field 1767). We restricted the sample to unrelated participants (kinship coefficient $<1/(2^{9/2})$)[76]. Second, our comparison sample

($N$ = 6500) was drawn at random from non-adopted participants who were unrelated to each other and to the adopted participants. Third, we identified 39,500 full siblings, excluding adopted individuals. We defined full-siblings as participants with a kinship coefficient between $1/(2^{(3/2)})$ and $1/(2^{(5/2)})$ and a probability of zero IBS sharing >0.0012, as suggested by[75] and[76].

After excluding the three sub-samples for polygenic score analyses and individuals related to these participants, we were left with 388,196 UK Biobank individuals with educational attainment (EA) data, and 202,815 individuals with cognitive performance (CP) data. We used these remaining individuals for the GWAS of EA and CP, and later meta-analysis with external GWASs[77] (see 'Statistical Analyses' and Supplementary Note 2).

**Twins Early Development Study (TEDS).** The Twins Early Development Study (TEDS) is a multivariate, longitudinal study of >10,000 twin pairs representative of England and Wales, recruited 1994–1996[78]. The demographic characteristics of TEDS participants and their families closely match those of families in the UK. Analyses were conducted on a sub-sample of dizygotic (DZ) twin pairs with genome-wide genotyping and phenotypic data on school achievement at age 12 (1431 DZ pairs) and age 16 (2398 pairs).

**Netherlands Twins Register (NTR).** The Netherlands Twin Register (NTR)[79] is established by the Department of Biological Psychology at the Vrije Universiteit Amsterdam and recruits children and adults twins for longitudinal research. Data on health, personality, lifestyle and others, as well as genotyping data have been collected on participants and their families.

We included in our analyses genotyped European-ancestry participants. We created a subsample of full-siblings. NTR contains information on numerous monozygotic multiples (twins or triplets). Because MZ multiples share the same genes, we randomly excluded all individuals but one per MZ multiple. Only siblings with complete genetic and outcome data were subsequently included in the analyses: 1631 siblings with CITO (achievement test taken during the last year of primary school) data (from 757 families) and 3163 siblings with EA data available (from 1309 families).

We created a subsample with complete offspring, maternal and paternal genotypic data (i.e., trios). Among individuals with available parental genotypes, respectively 1526 (from 765 families) and 2534 (from 1337 families) had reported CITO and EA information.

The sibling and trio subsets are not independent: for CITO, 823 participants are present in both subsets, 1374 for EA.

## Phenotypic measures

**UK Biobank.** Educational attainment and cognitive performance phenotypes were defined following Lee et al. 2018[77]. From data-field 6238, educational attainment was defined according to ISCED categories and coded as the number of Years of Education. The response categories are: none of the above (no qualifications) = 7 years of education; Certificate of Secondary Education (CSEs) or equivalent = 10 years; O levels/GCSEs or equivalent = 10 years; A levels/AS levels or equivalent = 13 years; other professional qualification = 15 years; National Vocational Qualification (NVQ) or Higher National Diploma (HNC) or equivalent = 19 years; college or university degree = 20 years of education. For cognitive performance, we used the (standardized) mean of the standardized scores of the fluid intelligence measure (data-field 20016 for in-person and 20191 for an online assessment).

**TEDS.** Educational achievement at age 12 was assessed by teacher reports, aggregated across the three core subjects (Mathematics, English, and Science).

Educational achievement at age 16 was assessed by self-reported results for standardized tests taken at the end of compulsory education in England, Wales and Northern Ireland: General Certificate of Secondary Education; GCSE). GCSE grades were coded from 4 (G; the minimum pass grade) to 11 (A∗; the highest possible grade). As with the age 12 measure, we analysed a variable representing mean score for the compulsory core subjects.

**NTR.** Educational attainment was measured by self-report of the highest obtained degree[80]. This measure was re-coded as the number of years in education, following Okbay et al. 2016[81].

Academic achievement is assessed in the Netherlands by a nationwide standardized educational performance test (CITO) around the age of 12 during the last year of primary education. CITO is used to determine tracking placement in secondary school in the Netherlands, in combination with teacher advice. The total score ranges from 500 to 550, reflecting the child's position relative to the other children taking the test this particular year.

## Genotype quality control

**UK Biobank.** SNPs from HapMap3 CEU (1,345,801 SNPs) were filtered out of the imputed UK Biobank dataset. We then did a pre-PCA QC on unrelated individuals, and filtered out SNPs with MAF < 0.01 and missingness > 0.05, leaving 1,252,123 SNPs. After removing individuals with non-European ancestry, we repeated the SNP QC on unrelated Europeans ($N$ = 312,927), excluding SNPs with MAF < 0.01, missingness > 0.05 and HWE $p < 10^{-10}$, leaving 1,246,531 SNPs. The HWE $p$-value threshold of $10^{-10}$ was based on: http://www.nealelab.is/blog/2019/9/17/genotyped-snps-in-uk-biobank-failing-hardy-weinberg-equilibrium-test. We then created a dataset of 1,246,531 QC-ed SNPs for 456,064 UKB subjects of European ancestry. Principal components were derived from a subset of 131,426 genotyped SNPs, pruned for LD ($r^2 > 0.2$) and long-range LD regions removed[82]. PCA was conducted on unrelated individuals using flashPCA v2[83].

**TEDS.** Two different genotyping platforms were used because genotyping was undertaken in two separate waves. AffymetrixGeneChip 6.0 SNP arrays were used to genotype 3665 individuals. Additionally, 8122 individuals (including 3607 DZ co-twin samples) were genotyped on Illumina HumanOmniExpressExome-8v1.2 arrays. After quality control, 635,269 SNPs remained for AffymetrixGeneChip 6.0 genotypes, and 559,772 SNPs for HumanOmniExpressExome genotypes.

Genotypes from the two platforms were separately phased and imputed into the Haplotype Reference Consortium (release 1.1) through the Sanger Imputation Service before merging. Genotypes from a total of 10,346 samples (including 3320 DZ twin pairs and 7026 unrelated individuals) passed quality control, including 3057 individuals genotyped on Affymetrix and 7289 individuals genotyped on Illumina. The identity-by-descent (IBD) between individuals was <0.05 for 99.5% in the merged sample excluding the DZ co-twins (range = 0.00–0.12) and ranged between 0.36 and 0.62 for the DZ twin pairs (mean = 0.49). There were 7,363,646 genotyped or well-imputed SNPs (for full genotype processing and quality control details, see[84]).

To ease high computational demands for the current study, we excluded SNPs with MAF < 1% and info <1. Following this, 619216 SNPs were included in polygenic score construction.

Principal components were derived from a subset of 39,353 common (MAF > 5%), perfectly imputed (info = 1) autosomal SNPs, after stringent pruning to remove markers in linkage disequilibrium ($r^2 > 0.1$) and excluding high linkage disequilibrium genomic regions to ensure that only genome-wide effects were detected.

**NTR.** Genotyping was done on multiple platforms, following manufacturers protocols: Perlegen-Affymetrix, Affymetrix 6.0, Affymetrix Axiom, Illumina Human Quad Bead 660, Illumina Omni 1 M and

Illumina GSA. For each genotype platform, samples were removed if DNA sex did not match the expected phenotype, if the PLINK heterozygosity F statistic was < −0.10 or >0.10, or if the genotyping call rate was <0.90. SNPs were excluded if the MAF < $1 \times 10^{-6}$, if the Hardy-Weinberg equilibrium p-value was <$1 \times 10^{-6}$, and/or if the call rate was <0.95. The genotype data was then aligned with the 1000 Genomes reference panel using the HRC and 1000 Genomes checking tool, testing and filtering for SNPs with allele frequency differences larger than 0.20 as compared to the CEU population, palindromic SNPs and DNA strand issues. The data of the different platforms was then merged into a single dataset, and one platform was chosen for each individual. Based on the -10.8 k SNPs that all platforms have in common, DNA identity-by-descent state was estimated for all individual pairs using the Plink 1.9 and King 2.1.6 programs. Samples were excluded if these estimates did not correspond to expected familial relationships. CEU population outliers, based on per platform 1000 Genomes PC projection with the Smartpca software v2.r904, were removed from the data. Then, per platform, the data was phased using Eagle v2.4.1 and then imputed to 1000 Genomes and Topmed using Minimac3-omp v2.10 following the Michigan imputation server protocols. Post-imputation, the resulting separate platform VCF files were merged with Bcftools 1.9 into a single file per chromosome for each reference, for SNPs present on all platforms. For the polygenic scoring and parental re-phasing, the imputed data were converted to best guess data and were filtered to include only ACGT SNPs, SNPs with MAF > 0.01, HWE $p > 10^{-5}$ and a genotype call rate >0.98, and to exclude SNPs with more than 2 alleles. All mendelian errors were set to missing. The remaining SNPs represent the transmitted alleles dataset. 20 PCs were calculated with Smartpca using LD-pruned 1000 Genomes−imputed SNPs genotyped on at least one platform, having MAF > 0.05 and not present in the long-range LD regions. Using the−tucc option of the Plink 1.07 software pseudo-controls for each offspring were created, given the genotype data of their parents. This resulted in the non-transmitted alleles dataset, as these pseudo-controls correspond to the child's non-transmitted alleles. To determine the parental origin of each allele, the transmitted and non-transmitted datasets were phased using the duoHMM option of the ShapeIT software. The phased datasets were then split based on parental origin, resulting in a paternal and maternal haploid dataset for the transmitted and non-transmitted alleles.

### Statistical analyses
All statistical tests are two-sided, unless otherwise stated.

**NonCog GWAS-by-subtraction.** To generate NonCog summary statistics, we implemented a GWAS-by-subtraction using Genomic SEM following Demange et al. 2020 using summary statistics of EA and cognitive performance obtained in samples independent from our polygenic score samples.

We ran a GWAS of Educational Attainment and Cognitive Performance in UK Biobank (polygenic score sample left-out). We meta-analysed them with the EA GWAS by Lee et al. excluding 23andMe, UK Biobank and NTR cohorts, and with the CP GWAS by Trampush et al. respectively (EA total N = 707,112 and CP N = 238,113) using Metal software release 2011-03-05. More information on these methods and intermediate GWAS are found in Supplementary Note 2 and Supplementary Data 1.

Following Demange et al. 2020, we used EA and CP meta-analysed summary statistics with GenomicSEM to create two independent latent variables: Cog, representing the genetic variance shared between EA and CP, and NonCog representing the residual genetic variance of EA when regressing out CP (Supplementary Fig. 1). These two latent factors were regressed on each SNP: we obtained association for 1,071,804 SNPs (HapMap3 SNPs, as recommended when comparing PGS analyses across cohorts). We calculate the

effective sample size of these GWAS to be 458,211 for NonCog and 223,819 for Cog.

**Polygenic Score construction in UK Biobank, TEDS and NTR.** Polygenic scores of NonCog and Cog were computed with Plink software (version 1.9 for NTR, 2 for UKB and TEDS)[85,86] based on weighted betas obtained using the LDpred v1.0.0 software using infinitesimal prior, a LD pruning window of 250 kb and 1000Genomes phase 3 CEU population as LD reference. Weighted betas were computed in a shared pipeline. In NTR, scores for non-transmitted and transmitted genotypes were obtained for fathers and mothers separately so we average them to obtain the mid-parent score.

### Polygenic score model fitting
Each model included cognitive and non-cognitive polygenic scores simultaneously and controlled for: 10 ancestry principal components (PCs), sex and age, interaction between sex and age, and cohort-specific platform covariate (NTR: genotyping platform, UKB: array, TEDS: batch). Age was estimated by year of birth, age at recruitment or age at testing depending on the cohorts, see Supplementary Data 2. Correlations between NonCog and Cog PGS, as well as between and within-family PGS are reported Supplementary Data 10.

Outcomes were standardized for each analysis group. Polygenic scores were standardised as follows prior to analysis. For the non-transmitted allele design, we summed the parental PGS and then scaled the non-transmitted and transmitted PGS separately, following Kong et al[26]. Note that the variances of the non-transmitted and transmitted PGS were not significantly different prior to scaling (Cog PGS: F = 1.0088, p = 0.71; NonCog PGS: F = 0.9920, p = 0.73). For the adoption design, we scaled the PGS in adopted and non-adopted groups separately. There were no significant differences in variances of adopted and non-adopted PGS prior to scaling (see Supplementary Data 11). For the sibling design, we scaled the PGS to have mean 0 SD 1 using the sibling group, and subsequently created the within-sibling PGS."

All regressions were linear models with lm() in R rather than mixed models as in previous analyses[27,28] and our pre-registered methods. See Supplementary Note 1 for the justification based on simulated data. We obtained bootstrapped standard errors and bias-corrected confidence intervals (normal approximation) for the population, indirect and direct effects, as well as the ratios of indirect/direct and indirect/population effect. We ran ordinary non-parametric bootstraps using 10,000 replications with boot() in R. For the sibling design, where two independent regressions are used, we use the same bootstrap samples for both (both regressions were run within the same boot object). For the adoption design, the bootstrapped samples are drawn from the adopted and non-adopted samples separately. The bootstrap estimates were used to test for the difference between the direct and indirect effect in both Cog and NonCog and the difference between the ratio indirect/population for Cog and NonCog, using Z-tests.

### Additional analyses (not pre-registered)
**Meta-analyses.** To estimate the overall indirect and direct effects of NonCog and Cog polygenic scores, we meta-analysed estimates across cohorts, designs and phenotypic outcomes.

To compare results obtained across the three different designs, we meta-analysed effect sizes obtained from each design across cohorts, but holding the outcome constant (educational attainment). The adoption design was only applied to EA in UKB, hence no meta-analysis was necessary.

Meta-analyses were conducted using the command rma.mv() in the R package metafor. Design was specified as a random intercept factor, except when results were meta-analysed within-design.

## Investigation of biases

**Population stratification.** Population stratification refers to the presence of systematic difference in allele frequencies across subpopulations, arising from ancestry difference due to non-random mating and genetic drift. This leads to confounding in genetic association studies. In a PGS analysis, bias due to population stratification can arise from both the GWAS used to create the scores and the target sample. We corrected for population stratification in the target sample by adjusting analyses for PCs (although this may not remove fine-scale stratification). For the GWAS summary statistics, the ratio statistics LDSC output is a standard measure of population stratification[87]. As a rule of thumb, an LDSC intercept higher than 1 (inflated) indicates presence of population stratification. Because we corrected the standard errors of the EA GWAS for inflation and GenomicSEM corrects for inflation as well, the ratio statistics of the Cog and NonCog GWAS are not a valid indication of population stratification (ratio < 0 following GC correction). We therefore use the ratio statistics of uncorrected EA and CP GWAS as proxies. Ratio and LDscore intercept was assessed with the ldsc software[87].

**Assortative mating.** Assortative mating refers to the non-random mate choice, with a preference for spouses with similar phenotypes. If these preferred phenotypes have a genetic component, assortative mating leads to an increased genetic correlation between spouses, as well as between relatives[52]. Assortative mating can therefore be inferred from elevated correlations between polygenic scores in siblings (correlations would be 0.5 without assortative mating) and between parents (correlations would be 0 without assortative mating). We estimated sibling intraclass correlations of Cog and NonCog PGS in UKB, TEDS and NTR, and Pearson's correlations of paternal and maternal Cog and NonCog PGS in NTR. Notably, these observed correlations cannot distinguish assortative mating from population stratification.

**Sibling effects.** We performed three additional analyses to investigate indirect genetic effects of siblings on educational outcomes.

First, we ran a linear mixed model extending our main non-transmitted alleles design to include polygenic scores of siblings (Eq. (4)). To this end, we used data from NTR on DZ pairs and both of their parents. Sample sizes of genotyped 'quads' with offspring CITO or EA phenotypes were 657 and 788, respectively.

$$
\begin{aligned}
EA = \ & \alpha_0 0 + \beta_{T\,Cog}(PGS(Cog)_T) + \beta_{T\,NonCog}(PGS(NonCog)_T) \\
& + \beta_{NT\,Cog}(PGS(Cog)_{NT}) + \beta_{NT\,NonCog}(PGS(NonCog)_{NT}) \\
& + \beta_{Sibling_{Cog}}\Big(PGS(Cog)_{Sibling}\Big) + \beta_{Sibling_{NonCog}}\Big(PGS(NonCog)_{Sibling}\Big) \\
& + sex + age + sex*age + 10PCs + genotyping\ platform
\end{aligned}
\tag{4}
$$

Second, we can also assess the presence of sibling genetic effects using monozygotic and dizygotic twins. Because monozygotic twins have the same genotypes, the genetically mediated environment provided by the cotwin is more correlated to the twin genotype in MZ twins than in DZ twins. The sibling genetic effect is more strongly reflected in the polygenic score prediction of the educational outcome for MZ twins than for DZ twins. If the sibling genetic effect is negative, the polygenic score effect (betas) on the outcome in people that have an MZ twin will be lower than in people that have a DZ twin, it will be higher in those with an MZ twin then those with an DZ twin if the sibling genetic effect is positive. We therefore compare Betas from Eq. (2) in a subset of MZ twins and in a subset of DZ twins (one individual per pair) in both NTR ($N_{MZ}$ = 818 & $N_{DZ}$ = 865 for CITO and $N_{MZ}$ = 1600 & $N_{DZ}$ = 1369 for EA) and TEDS ($N_{MZ}$ = 546 & $N_{DZ}$ = 2709).

Third, the presence of sibling genetic effects can be assessed using data on the number of siblings participants have. If an individual has more siblings, we expect their polygenic scores to be more correlated to sibling effects. As the number of siblings increases (if we assume linear increase) so does the degree to which a PGS captures sibling effects. If the sibling genetic effect is positive, the effect of the Cog and NonCog PGS on the educational outcome should increase with the number of siblings. However, family characteristics and environment might differ across families depending on the number of children. Therefore, changes in the effect of the PGS on our outcome with the number of siblings could be due to factors other than sibling genetic effects (for example, there is a known negative genetic association between number of children and EA[88] which could result in confounding). By also looking at changes in the effect of the Cog and NonCog PGS on the educational outcome in adopted (unrelated) sibships, we break the correlation between PGS and any sibling effects. If there is a change in PGS effect on the educational outcome in adopted children dependent on the number of (non-biological) siblings, we can assume this effect to be caused by mechanisms other than a sibling effect. We finally contrast the change in PGS depending on family size in biological and adopted siblings to get an idea of the sibling effect minus any other confounding effects of family size. We use the total number of reported siblings (full siblings for non-adopted and adopted siblings for adopted individuals, data-fields: 1873, 1883, 3972 & 3982).

### Reporting summary
Further information on research design is available in the Nature Research Reporting Summary linked to this article.

## Data availability
For the original summary statistics of Cog and NonCog, including NTR and UKBiobank siblings data, see[20]. The summary statistics for Cog and NonCog generated for this study are available at: https://doi.org/10.34894/MMXYPL. For UK Biobank dataset access, see: https://www.ukbiobank.ac.uk/using-the-resource/. Netherlands Twin Register data may be accessed, upon approval of the data access committee, email: ntr.datamanagement.fgb@vu.nl. Researchers can apply for access to TEDS data: https://www.teds.ac.uk/researchers/teds-data-access-policy.

## Code availability
All scripts used to run the analyses (empirical and simulated) are available at our GitHub https://github.com/PerlineDemange/GeneticNurtureNonCog/, which can be cited as Demange P., et al. Estimating effects of parents' cognitive and non-cognitive skills on offspring education using polygenic scores, GitHub, https://doi.org/10.5281/zenodo.6581326, 2022. All additional software used to perform the analyses are available online. The pre-registration of the study is available on OSF: https://osf.io/mk938/.

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

## Acknowledgements

We thank Dr. Aysu Okbay, the SSGAC and COGENT consortiums for sharing their summary statistics for GWAS of educational attainment and cognitive performance excluding specific cohorts. P.A.D. is supported by the grant 531003014 from The Netherlands Organisation for Health Research and Development (ZonMW). R.C. and E.M.E. are supported by the Research Council of Norway (288083). A.A. is supported by the Foundation Volksbond Rotterdam and by ZonMw grant 849200011 from The Netherlands Organisation for Health Research and Development. K.R. is supported the Wellcome Trust (213514/Z/18/Z). D.I.B. is supported by the Royal Netherlands Academy of Science (KNAW) Professor Award (PAH/6635). E.v.B. is supported by ZonMW grant 531003014 and VENI grant 451-15-017. M.G.N. is supported by R01MH120219, ZonMW grants 849200011 and 531003014 from The Netherlands Organisation for Health Research and Development, a VENI grant awarded by NWO (VI.Veni.191 G.030) and is a Jacobs Foundation Research Fellow. This research has been conducted using the UK Biobank Resource under Application Number 40310. The Netherlands Twin Register is supported by NWO Groot (480-15-001/674): Netherlands Twin Register Repository: researching the interplay between genome and environment and the Avera Institute for Human Genetics, Sioux Falls, South Dakota (USA) for genotyping. We gratefully acknowledge the research program 'Consortium on Individual Development (CID)' which is funded through the Gravitation program of the Dutch Ministry of Education, Culture and Science and the Netherlands Organization for Scientific Research (NWO: 0240-001-003). We gratefully acknowledge 'Open Data Infrastructure for Social Science and Economic Innovations (ODISSEI) (NWO: NRGWI.obrug.2018.008)'. We gratefully acknowledge the ongoing contribution of the participants in the Twins Early Development Study (TEDS) and their families. TEDS is supported by a program grant from the UK Medical Research Council (MR/V012878/1 and previously MR/M021475/1), with additional support from the US National Institutes of Health (AG046938). The funders had no role in study design, data collection and analysis, decision to publish or preparation of the manuscript.

## Author contributions

R.C. & P.A.D. conceived and designed the study, with helpful contributions from M.G.N. P.A.D. & R.C. analysed the data, with help from J.J.H. to obtain polygenic score weights and A.A. to perform GWAS in UK Biobank. P.A.D., M.G.N., R.C., and E.M.E. performed the simulation study. R.C. & P.A.D. wrote the manuscript. J.J.H., A.A., E.M.E., M.M., B.W.D., E.A.C., E.L.d.Z., K.R., D.I.B., E.v.B., and G.B. contributed to the interpretation of data, provided feedback on manuscript drafts, and approved the final draft.

## Competing interests

The authors declare no competing interests.
