## [Peer Review File · Nature Communications]

Estimating effects of parents' cognitive and non-cognitive skills on offspring education using polygenic scoresEditorial Note: This manuscript has been previously reviewed at another journal that is not operating a transparent peer review scheme. This document only contains reviewer comments and rebuttal letters for versions considered at *Nature Communications* .

REVIEWER COMMENTS

Reviewer #1 (Remarks to the Author):

I appreciate the additional analyses and revised text provided by the authors. It helped clear up my confusion in a lot of cases and has improved the manuscript. In particular, I appreciated the authors' additions clarifying how "noncognitive/not-cognitive skills" were defined. I was not totally satisfied by the authors' responses to two of my concerns, however. I describe these points below:

Major points:

1) I'm still quite confused by the simulations. Some of the results are puzzling to me, and as I'm trying to read the supplementary note to understand what was actually done and how to interpret the results, I'm having a hard time understanding the information provided. From what is written, this is what I'm understanding (along with several questions that I wasn't able to figure out by reading the Supplementary Note):

a) The authors simulate true effect sizes for 100 SNPs. They also simulate sibling effect sizes for the effect of a sibling's genotype on the proband? (Presumably, they also simulate the effect sizes for parental genotypes on the child's phenotype, but I couldn't find that information. I also wasn't sure where the simulated effect sizes come from for any of these groups. Are they drawn from a normal distribution or something? The authors say that they give all the sibling effects that same magnitude, so does that mean the effect of each sibling's SNP on the proband's phenotype is the same for all SNPs?)

b) The authors simulate 100 genotypes for 40,000 individuals. If there is population stratification, parents are assigned to different populations and different allele frequencies are used in each population. (Are there non-genetic phenotypic differences between the populations too? That is generally considered a large source of bias due to stratification.)

c) These individuals are paired up to make 20,000 parent pairs. If the simulation is meant to capture assortative mating, sorting is simulated by generating a sorting "phenotype" for them (equal to the PGS plus noise) and then match couples according to their rank.

d) The genomes for two biological children (the proband and the sibling) are simulated based on the genomes of the parents and one adopted child is simulated from a different set of parents simulated using a process similar to b and c.

e) The phenotypes were then simulated for the probands based on the effect sizes generated in step a and depending on the specific scenario being simulated. (E.g., if no sibling effects are assumed, then the effect of each sibling genotype would be zero.)

f) Proband, sibling, and parental PGSs are simulated by adding noise to the simulated effect sizes from step a and using these as PGS weights. (Were the same weights used for the siblings and parents as for the proband?)

g) Direct and indirect effect estimates were produced using the methods described in the paper.

Is this all correct? If so, I have the following questions:

i) The authors state that the reason that there is bias due to assortative mating is because of trans correlations between that aren't controlled for in the GWAS. However, in the simulation strategy above, there is no GWAS step and they just create PGSs based on the true underlying effect sizes (plus noise). In such a case, where does the bias enter? As I try to write out the model corresponding

to the simulation, I am unable to analytically identify where the bias would come from.

ii) In the population stratification simulation, if adopted children are always adopted within their sub-population (which I believe is what the authors report in Figure 3), I would have expected large biases of the indirect effects since the adoptive parents genotypes are predictive of their sub-population. Perhaps the reason for little bias in this case is that there aren't phenotypic differences between the populations. For these simulations to more realistically reflect population stratification, the authors should probably simulate both allele frequency and phenotypic differences between the sub-populations.

2) I appreciate the analyses the authors did to assess whether birthweight could be driving the estimates of the parental genetic effects. I believe I did a poor job explaining my concern though. My concern is that there appear to be real, substantial differences (as large as 0.3 standard deviations) between the adopted and nonadopted samples. While the authors find that controlling for reported birthweight has little effect on the results they report, this doesn't resolve my concern for two reasons. First, if birthweight is measured with error, then it is only an imperfect control. Second, and most importantly, if there are substantial differences in observed variables such as birthweight, this makes me very nervous about unobserved ways in which the adoptees differ from adoptees. This doesn't bother me very much for the estimates of the direct genetic effects since you can specify that these are direct effects for the adoptee population. But for the indirect genetic effect estimates, you must assume that the adopted and non-adopted samples are comparable. Otherwise, I don't think that the differences in the estimates has a meaningful interpretation for any population. I do not see how this concern can be resolved other than by making it clear that this is a severe limitation of estimating indirect parental effects by the approach proposed by the authors and that these estimates are unlikely to be reliable. (To be honest, making these limitations clear could be an important contribution to the literature by this paper, assuming the authors agree with my critique.)

Minor point:

3) I'm not totally sure how to fix this, but Figure 3 is very difficult to read. There is a lot of text on the x-axis, and it is often very distant from the bars associated with the x-axis labels. Maybe this could be resolved with some strategic shading? Or maybe with the different components listed on the left-hand-side and X's marking with components are included in each simulation?

Reviewer #4 (Remarks to the Author):

I appreciate the authors' detailed responses to my and the other reviewers' comments. I have no further suggestions.

Reviewer #1 (Remarks to the Author):

I appreciate the additional analyses and revised text provided by the authors. It helped clear up my confusion in a lot of cases and has improved the manuscript. In particular, I appreciated the authors' additions clarifying how "noncognitive/not-cognitive skills" were defined. I was not totally satisfied by the authors' responses to two of my concerns, however. I describe these points below:

Major points:

1) I'm still quite confused by the simulations. Some of the results are puzzling to me, and as I'm trying to read the supplementary note to understand what was actually done and how to interpret the results, I'm having a hard time understanding the information provided. From what is written, this is what I'm understanding (along with several questions that I wasn't able to figure out by reading the Supplementary Note):

- a) The authors simulate true effect sizes for 100 SNPs. They also simulate sibling effect sizes for the effect of a sibling's genotype on the proband? (Presumably, they also simulate the effect sizes for parental genotypes on the child's phenotype, but I couldn't find that information. I also wasn't sure where the simulated effect sizes come from for any of these groups. Are they drawn from a normal distribution or something? The authors say that they give all the sibling effects that same magnitude, so does that mean the effect of each sibling's SNP on the proband's phenotype is the same for all SNPs?)
- b) The authors simulate 100 genotypes for 40,000 individuals. If there is population stratification, parents are assigned to different populations and different allele frequencies are used in each population. (Are there non-genetic phenotypic differences between the populations too? That is generally considered a large source of bias due to stratification.)
- c) These individuals are paired up to make 20,000 parent pairs. If the simulation is meant to capture assortative mating, sorting is simulated by generating a sorting "phenotype" for them (equal to the PGS plus noise) and then match couples according to their rank.
- d) The genomes for two biological children (the proband and the sibling) are simulated based on the genomes of the parents and one adopted child is simulated from a different set of parents simulated using a process similar to b and c.
- e) The phenotypes were then simulated for the probands based on the effect sizes generated in step a and depending on the specific scenario being simulated. (E.g., if no sibling effects are assumed, then the effect of each sibling genotype would be zero.)
- f) Proband, sibling, and parental PGSs are simulated by adding noise to the simulated effect sizes from step a and using these as PGS weights. (Were the same weights used for the siblings and parents as for the proband?)
- g) Direct and indirect effect estimates were produced using the methods described in the paper.

Is this all correct? If so, I have the following questions:

We are grateful to the reviewer for engaging with the simulation study in detail and identifying parts requiring more explanation. The reviewer's description is mainly correct. We address specific questions below, and have also added details in the Supplementary Note.

- a) True SNP effects are drawn from a normal distribution. GWAS effects are just 'true' effects plus noise. All biases and components (parental indirect genetic effects, sibling indirect genetic effects etc.) except for population stratification are simulated to influence target population phenotypes, not the GWAS effects. For population stratification, we simulated the presence of subpopulations in the GWAS and target samples (see below). The 'equal magnitude of sibling effects' referred to offspring phenotypes having equal effects on one another (i.e., no birth order effects or different effects for adoptive siblings), not the equality of sibling effects for all SNPs.
- b) To simulate population stratification, we create two subpopulations with not only different allele frequencies but mean differences in phenotypes. We thank the reviewer for alerting us that this was unclear. We have expanded the description in the Supplementary Note (pg. 8): *'Population stratification can be conceptualized as systematic differences in allele frequencies between sub-populations. These frequency differences cause confounding in genetic studies when phenotypes also differ between sub-populations. We simulate such sub-populations in both the GWAS discovery and target PGS analyses samples. We first create new genotypes in two groups, drawing upon two different sets of simulated minor allele frequency distributions. We also define a phenotypic difference between these two groups, by including an 'environmental confounding' parameter for different noise and means for the GWAS phenotypes between subpopulations. We then run a single GWAS in these two populations. We create phenotypes and polygenic scores (based on the GWAS results) in a target sample of families, comprising the same two sub-populations present in the GWAS. Our simulation allows for adoptees to be matched with adoptive parents both within- and between- sub-populations. We report results from a simulation with adoptees matched with adoptive parents within the same sub-population.'*

e) (and a)) We use the true SNP effects (drawn from a normal distribution) to simulate 'true' genetic scores for mothers, fathers, and biological and adopted offspring. These true genetic scores are then used to make phenotypes. The true SNP effects are the same for all individuals and for all sub-populations. We have clarified this procedure in the Supplementary Note.

i) The authors state that the reason that there is bias due to assortative mating is because of trans correlations between that aren't controlled for in the GWAS. However, in the simulation strategy above, there is no GWAS step and they just create PGSs based on the true underlying effect sizes (plus noise). In such a case, where does the bias enter? As I try to write out the model corresponding to the simulation, I am unable to analytically identify where the bias would come from.

The reviewer is correct that we simulated assortative mating in the target sample, and not in the GWAS effects. We apologise for introducing confusion in our previous response by saying *'AM leads to trans correlations (i.e., between locus A in parent 1 and locus B in parent 2), inducing bias in the GWAS effect*

estimates of both the transmitted and non-transmitted alleles.' This was not relevant since it is not what we simulated.

The Supplementary Note (pg. 8) contains the following explanation of our simulation of assortative mating: *'To simulate assortment, we re-create offspring genotypes and polygenic scores after matching parents together systematically (instead of randomly as above). We first create phenotypes for the parents (based on true genetic score plus noise), rank the mothers and fathers by phenotype, and match couples according to rank (i.e., mothers with higher phenotypic values match with fathers with higher phenotypic values). Since mating does not perfectly track with phenotypic rank, we add noise to the ranking of mothers and fathers prior to matching, following a chosen phenotypic correlation. Offspring genotypes are then simulated as random draws from the matched couples' genotypes. Assortment is simulated to be the same strength for adoptees' and nonadoptees' parents, and we simulate random placement by un-ranking adoptees before matching them to adoptive families.'*

Therefore, the only step that introduces bias is the assortment of the parents in the target sample on a heritable phenotype. Although it is unrealistic to assume that the GWAS effects are not also biased by assortative mating, we simulated assortment only in the target population to make the results interpretable.

The Supplementary Note then describes the results of the assortment simulation and offers some potential explanation. Following comments of the reviewer, it became visible that noise in the GWAS SNP effects affects how assortment introduces bias, we therefore extended our Supplementary Note to describe this detail (and adjusted the manuscript): *'In our main scenario, which includes substantial error in the GWAS SNP effects used to calculate polygenic scores, so the correlation of GWAS SNP effects and true SNP effects is on average 0.45), we found that the bias from assortative mating in the indirect genetic effect estimate was lower in the adoption design than in the non-transmitted allele and sibling designs. We also tested other scenarios with lower error in the SNP effects used to make the polygenic score. In the scenario with assortative mating but not indirect effects, lower error in the polygenic scores led to decreased bias in estimates from the NT and sibling designs. In the scenario with both assortative mating and indirect effects, with decreasing error in SNP effects, the sibling estimate is consistently biased, but the adoption estimate is more biased and the NT estimate is less biased. Results of other simulations did not change according to the error. We present in the main manuscript the initial results with substantial error as the most conservative and realistic example. In real data, we expect this bias due to the combination of error in effect sizes and non-random mating to decrease as GWAS sample sizes increases.*

Bias in the sibling design likely arises as the population effect contains assortative mating while the within-sibling effect does not. Bias in the non-transmitted allele design due to assortative mating, which happens due to correlations between parental alleles, is described in Kong et al. 2018 (Kong et al. 2018). Interestingly, the bias in the adoption design from assortative mating is zero in the absence of a parental indirect genetic effect, but slightly above zero when a parental indirect genetic effect was also specified. In other words, the presence of parental indirect genetic effects is required for assortative mating to bias

estimates from the adoption design. We simulated the same strength of assortative mating for the parents of both adopted and non-adopted individuals, so the result cannot be due to elevated assortment in the latter group (leading to residual assortment in the indirect effect estimate when calculating $\beta_{\text{non-adopted}} - \beta_{\text{adopted}}$). Such differences could exist in the real data, but there is scarce and inconsistent evidence regarding assortment in biological parents of adoptees versus other parents (Plomin et al. 1977; Ho et al. 1979). Overall, the results suggest that assortative mating could explain lower estimates of indirect genetic effects from the adoption design compared to the other designs, but may depend on the level of noise in the GWAS effects.'

ii) In the population stratification simulation, if adopted children are always adopted within their sub-population (which I believe is what the authors report in Figure 3), I would have expected large biases of the indirect effects since the adoptive parents genotypes are predictive of their sub-population. Perhaps the reason for little bias in this case is that there aren't phenotypic differences between the populations. For these simulations to more realistically reflect population stratification, the authors should probably simulate both allele frequency and phenotypic differences between the sub-populations.

The reviewer is right that we simulated a scenario where adopted children are adopted by parents who are part of the same sub-population as their biological parents. We agree that, due to this specification, adoptive parents' genotypes are predictive of their adopted children's sub-population. As a result, genetic effects estimated among adoptees will capture population stratification. In our simulation, adoptive parents' genotypes predict their non-adopted children's sub-population to the same extent as they predict adoptees' sub-population. So genetic effects estimated among non-adoptees will capture population stratification to the same extent. Hence, the bias is cancelled out when taking the difference between effects estimated in the two groups. If adoptions were random such that children and parents' ancestries were different (e.g., cross-continental adoption), we speculate that direct genetic effect estimates may be less biased by population stratification. Consequently, indirect genetic effects may be overestimated to the extent that population stratification affects the non-adopted sample. Population stratification in adoptive and non-adopted families is an interesting area for further research.

Since we did simulate both allele frequency and phenotypic differences between the sub-populations (described on Supplementary Note pg. 8), we do not think that the simulations are missing any source of bias associated with population stratification.

2) I appreciate the analyses the authors did to assess whether birthweight could be driving the estimates of the parental genetic effects. I believe I did a poor job explaining my concern though. My concern is that there appear to be real, substantial differences (as large as 0.3 standard deviations) between the adopted and nonadopted samples. While the authors find that controlling for reported birthweight has little effect on the results they report, this doesn't resolve my concern for two reasons. First, if birthweight is measured with error, then it is only an imperfect control. Second, and most importantly, if there are substantial differences in observed variables such as birthweight, this makes me very nervous about unobserved ways in which the adoptees differ from adoptees. This doesn't bother me very much

for the estimates of the direct genetic effects since you can specify that these are direct effects for the adoptee population. But for the indirect genetic effect estimates, you must assume that the adopted and non-adopted samples are comparable. Otherwise, I don't think that the differences in the estimates has a meaningful interpretation for any population. I do not see how this concern can be resolved other than by making it clear that this is a severe limitation of estimating indirect parental effects by the approach proposed by the authors and that these estimates are unlikely to be reliable. (To be honest, making these limitations clear could be an important contribution to the literature by this paper, assuming the authors agree with my critique.)

We thank the reviewer for clarifying their point. We agree that the magnitude of the difference in birthweight between adoptees and non-adopted individuals, while in the category of 'small' (<https://www.spss-tutorials.com/cohens-d/>), is substantial enough to be concerning. We have added text to highlight this finding, and what it means for estimating indirect genetic effects. We have also added an explicit statement of our assumptions when we introduce the adoption design (Results). In general, we have edited the whole manuscript to ensure that we do not claim that the adoption design has fewer problems than the other designs.

Results: *'the indirect genetic effect estimate based on the difference assumes that the two samples are the same in terms of all observed and unobserved confounders, an untestable assumption that is unlikely to always hold. We found small differences between adoptees and non-adopted individuals in the UK Biobank in their demographic and early-life characteristics. Cohen's d values were: $d < 0.15$ for Cog and NonCog PGS and educational attainment, and $d = 0.31$ for the birth weight.'*

Discussion: *'Given that there was no evidence for sibling effects of the Cog or NonCog PGS our adoption-based estimates, which appear to be less biased by population stratification and assortative mating, should provide a closer measure of parental indirect genetic effects, in the absence of other issues. However, adoptees and non-adopted individuals differ in unobserved and observed ways, including birthweight ($d = 0.3$). These differences likely make adoption-based estimates of indirect genetic effects, which rely on a comparison of the two groups, less reliable. Moreover, three factors may make the adoption-based estimates of indirect genetic effects too conservative. First, adoption based indirect effect estimates exclude prenatal indirect genetic effects (and indirect genetic effects taking place between the birth and moment of adoption), [...] Second, adoptees may have been exposed to a narrower range of environments (e.g., family socioeconomic status) compared to non-adopted individuals [...] Third, selective placement of children in adoptive families matching characteristics of their biological families, or adoption of children by close relatives 61, could result in correlation between child and (adoptive) parent genotypes, leading to an underestimation of the indirect genetic effect [...].'*

Supplementary Note: *'We observed differences in birthweight, with adopted individuals being lighter than non-adopted individuals (mean = 3.12kg vs 3.33kg, Cohen's $d = 0.31$). The variance in birthweight was also significantly different, with more variance in the adoptee group (0.57 vs 0.45). However, missingness of birthweight data is severe in both groups, but particularly adoptees (72% of missing data among adoptees, 43% in the non-adopted group).*

Overall, the key variables under study are well-matched between the adopted and non-adopted groups. There may be stronger systematic differences relating to birthweight and geography (as well as unobserved variables). Differences between the adopted and non-adopted groups are more of a concern for estimating indirect genetic effects than direct genetic effects, since the former only are based on comparison between groups. Direct genetic effects estimated using this design can be interpreted specifically for the adoption sample, whereas indirect genetic effect estimates have a more ambiguous interpretation since they are based on two different groups.'

Minor point:

3) I'm not totally sure how to fix this, but Figure 3 is very difficult to read. There is a lot of text on the x-axis, and it is often very distant from the bars associated with the x-axis labels. Maybe this could be resolved with some strategic shading? Or maybe with the different components listed on the left-hand-side and X's marking with components are included in each simulation?

To make Figure 3 easier to read, we have added shading behind every other set of bars, and numbering on the x-axis labels.

Reviewer #4 (Remarks to the Author):

I appreciate the authors' detailed responses to my and the other reviewers' comments. I have no further suggestions.

We thank the reviewer for the positive feedback.

REVIEWERS' COMMENTS

Reviewer #1 (Remarks to the Author):

I am satisfied with the author's revision. I have learned a lot from this paper and feel like it is a valuable contribution to the literature.

Reviewer #5 (Remarks to the Author):

Demange and colleagues are to be congratulated for well conceptualized analysis strategy, including pre-registration of key analyses, and the use of multiple genetic designs to triangulate on questions concerning the genetic influences on cognitive and non-cognitive abilities on educational attainment. While this study has a number of strengths, this reviewer is concerned by the intense reductionism, both explicit and implicit, present in the work. Educational attainment, as the authors note, is a complex trait influenced by many different factors including "innate" cognitive ability, personality factors, socioeconomic status and class, race, neurological or mental illness, and access, among others. From this myriad of possible influences, the authors previously demonstrated that using common genetic variants, one could dissociate cognitive abilities from non-cognitive abilities relative to their impact on education levels (Demange et al., Investigating the genetic architecture of noncognitive skills using GWAS-by-subtraction, *Nat Genet.* 2021 Jan;53(1):35-44.). In this formulation, the non-cognitive variable includes anything not associated with a set of cognitive measures, including measurement error. Thus, by definition, the non-cognitive variable is dependent upon the quality and completeness of the cognitive assessment, since any variance not captured by the specific cognitive tests employed will be included in the non-cognitive variable. More directly, the level of cognitive variance attributed to non-cognitive ability is necessary dependent on the cognitive assessment performed. Hence, this reviewer's assessment of Demange and colleague's experiment focused on examining and comparing the cognitive tests performed in the UK Biobank (UKB), TEDS and NTR cohorts. While the TEDS and NTR battery of assessments were reasonably comprehensive, the UKB assessment was rather limited and has notoriously poor test re-test reliability (Davies et al., Genome-wide association study of cognitive functions and educational attainment in UK Biobank (N=112,151), *Mol Psychiatry.* 2016 Jun;21(6):758-67). Based on this observation, one would anticipate differences between the UKB and other cohorts purely related to the particulars of cognitive assessment. Although the authors did observe such between cohort differences, the quality and completeness of the cognitive assessments was not discussed as a possible cause of these effects. The reductionism inherent in the logical process used to develop a single non-cognitive variable was also present when making decisions about the statistical genetic analyses. Rather than use classical variance components approaches, the authors focused on deriving polygenic scores. This decision has two limitations. First, it is unclear how much of the genetic variance is explained by the various PGS and if the variance explained is relatively stable across cohorts and samples. How much of the genetic variance need be explained by a PGS before the authors are comfortable using the index as a proxy of complete genetic variation (an implicit assumption of the manuscript)? Second, analysis choice implicitly requires that the cognitive and non-cognitive variables are each explained a single genetic factor. Yet, different cognitive abilities are thought to have both overlapping and unique genetic influences. This richness is lost when calculating a single PGS. This concern would appear to be intensified in the context of the non-cognitive variable, where ostensibly many different skills and factors are represented. If Demange and colleague's are under-representing genetic influences due to the use of PGSs, then the assessment of direct and indirect genetic effects becomes very difficult to interpret.

Although I have a number of additional concerns, I believe that these are the most fundamental.

Responses to reviewer 5 for the article 'Estimating effects of parents' cognitive and non-cognitive skills on offspring education using polygenic scores'

Demange and colleagues are to be congratulated for well conceptualized analysis strategy, including pre-registration of key analyses, and the use of multiple genetic designs to triangulate on questions concerning the genetic influences on cognitive and non-cognitive abilities on educational attainment. While this study has a number of strengths, this reviewer is concerned by the intense reductionism, both explicit and implicit, present in the work. Educational attainment, as the authors note, is a complex trait influenced by many different factors including "innate" cognitive ability, personality factors, socioeconomic status and class, race, neurological or mental illness, and access, among others. From this myriad of possible influences, the authors previously demonstrated that using common genetic variants, one could dissociate cognitive abilities from non-cognitive abilities relative to their impact on education levels (Demange et al., Investigating the genetic architecture of noncognitive skills using GWAS-by-subtraction, Nat Genet. 2021 Jan;53(1):35-44.). In this formulation, the non-cognitive variable includes anything not associated with a set of cognitive measures, including measurement error. Thus, by definition, the non-cognitive variable is dependent upon the quality and completeness of the cognitive assessment, since any variance not captured by the specific cognitive tests employed will be included in the non-cognitive variable.

We agree with the reviewer that the quality of the latent non-cognitive construct is dependent upon the cognitive assessments used for the discovery GWAS. We acknowledge this in the Discussion:

'Some cognitive skills might not be reflected in the available Cognitive Performance GWAS, so the NonCog factor could capture genetic influences affecting cognition. However, previous analyses have shown that a NonCog PGS based on GWAS-by-subtraction predicts substantially less variation in cognition than the Cog PGS⁶³.'

This issue is also discussed extensively in the original GWAS-by-subtraction paper (Demange et al. 2021, Nature Genetics).

More directly, the level of cognitive variance attributed to non-cognitive ability is necessary dependent on the cognitive assessment performed. Hence, this reviewer's assessment of Demange and colleague's experiment focused on examining and comparing the cognitive tests performed in the UK Biobank (UKB), TEDS and NTR cohorts. While the TEDS and NTR battery of assessments were reasonably comprehensive, the UKB assessment was rather limited and has notoriously poor test re-test reliability (Davies et al., Genome-wide association study of cognitive functions and educational attainment in UK Biobank (N=112,151), Mol Psychiatry. 2016 Jun;21(6):758-67). Based on this observation, one would anticipate differences between the UKB and other cohorts purely related to the particulars of cognitive assessment. Although the authors did observe such between cohort differences, the quality and completeness of the cognitive assessments was not discussed as a possible cause of these effects.

While the reviewer is right that the level of cognitive variance attributed to non-cognitive is dependent on the cognitive assessment performed for the discovery GWAS, it is not accurate to say that we examined and compared cognitive tests performed in UKB, TEDS, and NTR. We in fact compare education achievement and attainment outcomes.

The polygenic scores for cognitive and non-cognitive skills we derive in the three samples are based on the same cognitive assessment (the measure of the cognitive performance and educational attainment GWAS).

Therefore, differences in results between cohorts are not explained by differences in cognitive assessments. Cohort differences are more likely to be explained by whether achievement or attainment was measured, and by national/cultural differences. We discuss the such differences in depth in the article, for example:

'The lower importance of parental indirect genetic effects for child achievement in the Netherlands compared to the UK indicates that our UK achievement outcomes more strongly capture variation in family background. This difference could result from the design of these achievement measures: Dutch test results are standardized based on a representative population, but UK teacher reports might still be affected by student social background... For adult attainment, results were more similar across UK and Dutch cohorts, corresponding with recent evidence for consistent shared-environment influence on educational attainment across social models.'

The reductionism inherent in the logical process used to develop a single non-cognitive variable was also present when making decisions about the statistical genetic analyses. Rather than use classical variance components approaches, the authors focused on deriving polygenic scores. This decision has two limitations. First, it is unclear how much of the genetic variance is explained by the various PGS and if the variance explained is relatively stable across cohorts and samples. How much of the genetic variance need be explained by a PGS before the authors are comfortable using the index as a proxy of complete genetic variation (an implicit assumption of the manuscript)?

By using PGS, we obtain indices of non-cognitive and cognitive skills that are the same across all cohorts (i.e., compared to relying on various phenotypic measures of skills). This allows us to explicitly compare genetic effects across educational outcomes in UKB, TEDS and NTR – i.e., in contrast to the reviewer's first point, the variance explained by PGS for each cohort is reported in our article. It is currently unfeasible to estimate full variance components for parental indirect genetic effects, since we are not aware of any extended twin-family datasets with measured non-cognitive skills.

We acknowledge in the Discussion that the use of PGS means that we only capture a fraction of the genetic influence on education (therefore less than twin studies). This is a limitation of all PGS studies. Our findings of indirect genetic effects are all the more striking given the lower variance explained by PGS than by twin-based variance components.

'The environmental effects of parents' non-cognitive and cognitive skills are likely to be larger than we estimate, because our approach only captures effects of parent skills tagged by current GWAS. Polygenic scores index a subset of the common genetic component of parent skills, which is in turn a fraction of the total genetic component (missing heritability^{44,45}), and cannot account for the non-heritable component of parent skills.'

Second, analysis choice implicitly requires that the cognitive and non-cognitive variables are each explained a single genetic factor. Yet, different cognitive abilities are thought to have both overlapping and unique genetic influences. This richness is lots when calculating a single PGS. This concern would appear to be intensified in the context of the non-cognitive variable, where ostensibly many different skills and factors are represented. If Demange and colleague's are under-representing genetic influences due to the use of PGSs, then the assessment of direct and indirect genetic effects becomes very difficult to interpret.

Although I have a number of additional concerns, I believe that these are the most fundamental.

We thank the reviewer for this useful point. We agree that different non-cognitive skills have overlapping and distinct genetic influences, and we are only focussing on the genetic variance in common between them. Whilst creating an overall non-cognitive measure that is agnostic to specific measures is a key aim of our study, we acknowledge that it will be important in future to investigate the roles of specific skills in parental effects. In the Discussion, we write:

'It may be useful to develop a more precise non-cognitive skills GWAS, by creating the latent Cog and NonCog factors using additional measured phenotypes. To this end, large GWA meta-analyses should be completed not only for personality⁶⁷ but not for other traditional non-cognitive skills such as motivation and self-control.'